# Development and Comparison of Treatment Decision Tools for Glucocorticoid-Induced Osteoporosis

**DOI:** 10.3390/diagnostics14040452

**Published:** 2024-02-19

**Authors:** Jia-Feng Chen, Shan-Fu Yu, Wen-Chan Chiu, Chi-Hua Ko, Chung-Yuan Hsu, Han-Ming Lai, Ying-Chou Chen, Yu-Jih Su, Hong-Yo Kang, Tien-Tsai Cheng

**Affiliations:** 1Division of Rheumatology, Allergy, and Immunology, Department of Internal Medicine, College of Medicine, Kaohsiung Chang Gung Memorial Hospital, Chang Gung University, Kaohsiung 833, Taiwan; uporchidjfc@gmail.com (J.-F.C.); 2Graduate Institute of Clinical Medical Sciences, College of Medicine, Chang Gung University, Kaohsiung 833, Taiwan; 3Center for Menopause and Reproductive Medicine Research, Department of Obstetrics and Gynecology, College of Medicine, Kaohsiung Chang Gung Memorial Hospital, Chang Gung University, Kaohsiung 833, Taiwan; 4Division of Endocrinology and Metabolism, Department of Internal Medicine, College of Medicine, Kaohsiung Chang Gung Memorial Hospital, Chang Gung University, Kaohsiung 833, Taiwan; 5Department of Biological Sciences, National Sun Yat-Sen University, Kaohsiung 804, Taiwan

**Keywords:** corticosteroids, fracture risk assessment, osteoporosis, screening, fracture prevention, treatment

## Abstract

Long-term Glucocorticoid (GC) use results in compromised bone strength and fractures, and several treatment recommendations have been developed to prevent fractures, but none have been validated in a real-world setting. This study aims to create a treatment decision tool and compares this tool to the treatment suggestions from the American College of Rheumatology (ACR), International Osteoporosis Foundation and European Calcified Tissue Society (IOF-ECTS), and GC-adjusted Fracture Risk Assessment Tool (GC-FRAX), above the intervention threshold. We utilized registry data gathered at Chang Gung Memorial Hospital in Kaohsiung, Taiwan, between September 2015 and April 2021. This research is a single-center, observational, and case-controlled study. We recruited participants using prednisone for at least 2.5 mg/day or the equivalent dose for over 3 months, excluding those younger than 40, those with malignancies, or those currently undergoing anti-osteoporosis therapy. The primary endpoint was new fragility fractures within 3 years, including morphometric vertebral fractures detected at baseline and with a follow-up thoracic–lumbar spine X-ray. Participants were randomly allocated into derivation and validation sets. We developed the Steroid-Associated Fracture Evaluation (SAFE) tool in the derivation cohort by assessing the weights of exploratory variables via logistic regression. Prediction performance was compared in the validation set by the receiver operating characteristic (ROC) curve, the area under the curve (AUC), and sensitivity and specificity. A total of 424 treatment-naïve subjects were enrolled, and 83 (19.6%) experienced new fractures within 3 years. The final formula of the SAFE tool includes osteoporosis (1 point), an accumulated GC dose ≥ 750 mg within 6 months (or equivalent prednisolone of ≥4.5 mg/day for 6 months) (1 point), a BMI ≥ 23.5 (1 point), previous fractures (1 point), and elderliness of ≥70 years (2 points). In the validation set, a treatment decision based on the SAFE ≥ 2 points demonstrated an AUC of 0.65, with a sensitivity/specificity/accuracy of 75.9/54.0/58.9, with an ACR of 0.56 (100.0/11.0/31.0), IOF-ECTS 0.61 (75.9/46.0/52.7), and GC-FRAX 0.62 (82.8/42.0/51.2). Among current GIOP recommendations, the SAFE score serves as an appropriate treatment decision tool with increased accuracy and specificity.

## 1. Introduction

Glucocorticoids (GCs), prevalent anti-inflammatory drugs, have been broadly used to treat connective tissue diseases. The prevalence of oral GC usage ranges from 0.9 to 1.2% in various populations, increasing to 2.5~3.1% in those over 70 years old [1,2,3,4]. Long-term GC administration is related to a high prevalence of osteoporosis or fracture, ranging from 21% to 30% in previous literature reviews [5]. Osteoporosis or fragility fracture has been the most frequently studied and concerning adverse event in patients on long-term GCs [6,7,8]. In a longitudinal observational study of patients with rheumatoid arthritis, osteoporotic fractures increased in a dose-dependent manner, from a hazard ratio of 1.26 (GC dose < 7.5 mg/day) to 1.57 (≥7.5 mg/day) [9]. The fracture risk was approximately twice as high as those without GC exposure [10]. Moreover, the fracture rate may be underestimated due to the high prevalence of asymptomatic vertebral fractures [11]. Glucocorticoid-induced osteoporosis (GIOP) results in a substantial economic burden in modern societies, estimated to be from USD 1743 to USD 18,358 per event in 2009 [12].

Doubtlessly, high doses of GCs should be avoided to reduce bone-destructive effects. Low doses of GCs with the shortest duration possible is suggested as an initial or bridge therapy by the American College of Rheumatology/European Alliance of Associations for Rheumatology (ACR/EULAR) guidelines [13,14]. Nevertheless, due to their low cost and profound anti-inflammatory effects on controlling disease flares, GCs are still widely used for extended periods. The long-term observational study of TOMORROW (Total Management of Risk Factors in Rheumatoid Arthritis Patients to Lower Morbidity and Mortality) discovered that even low doses of GCs increased clinical fracture risks [15]. The GLORIA (Glucocorticoid Low-Dose in Rheumatoid Arthritis) study, a double-blind, randomized trial comparing low-dose GCs with a placebo, revealed a significant decrease in lumbar spine bone mineral density (BMD) and a non-significant increase in new compression fractures after a 2-year follow-up [16]. The early recognition of GIOP in patients and subsequent prescriptions of approved anti-osteoporotic therapy (AOT) has become a vital issue in restoring musculoskeletal health and reducing medical costs.

Several assessment criteria were considered therapeutic thresholds for GIOP, such as bone mineral density (BMD) and the Fracture Risk Assessment Tool (FRAX). BMD changes may be subtle in GIOP, and BMD thresholds for the prevention of GIOP fractures are variable and debated [17]. Although a GC-adjusted FRAX score (GC-FRAX) is strongly associated with fracture rate, an intervention threshold is required to determine which patients need treatment. However, there is no universal consensus on the intervention threshold for GIOP. Although country-specific intervention thresholds assessed by FRAX, presented as a fixed or age-dependent value, are widely used in postmenopausal women [18], GC-FRAX-based intervention thresholds require further validation. Other treatment recommendations are more complicated, incorporating multiple convention risk factors in combination with the GC-FRAX intervention threshold [19], such as ACR [20] and the International Osteoporosis Foundation and European Calcified Tissue Society (IOF-ECTS, Appendix A) [21]. These suggestions are predominantly decided on by the systemic review of published randomized controlled trials and experts’ consensus [19] rather than the clinical database. This research aimed to compare the performance of different treatment recommendations and develop a scoring criterion based on a real-world database, assisting general practitioners in identifying high-fracture-risk patients and treatment decision making.

## 2. Materials and Methods

### 2.1. Study Participants

A registry was compiled at Chang Gung Memorial Hospital, Kaohsiung (CGMHK), Taiwan, between September 2015 and April 2021, aiming to track BMD changes and fractures in patients with connective tissue diseases, as described in a previous work [22]. Adults with RA, lupus, Sjögren’s syndrome, vasculitis, systemic sclerosis, allergy, and inflammatory myopathy were confirmed by licensed rheumatologists. We selected participants who were currently or had been exposed to GCs for over 3 months of at least 2.5 mg/day of prednisolone or equivalents, including methylprednisolone, cortisol, and dexamethasone, via administration of oral, intravenous, or intramuscular routes. The accumulated GC dose was calculated at 3 and 6 before the index day. We excluded participants who were younger than 40 years of age, had any malignancies within the previous 5 years, or used AOT, including bisphosphonates, selective estrogen-receptor modulators, denosumab, or teriparatide on the index day. Participants exposed to AOT within 12 months before the index day or who used AOT within 6 months after the index day were excluded to reduce the bias of medication effects that may have protected patients from fractures. Patients’ BMD was measured for the femoral neck, total hip, and lumbar spine (L1–L4) using a dual-energy X-ray absorptiometry scanner (Delphi A; Hologic Corp., Waltham, MA, USA). Body height, weight, body mass index, and fracture risk factors were documented at enrollment. All fracture types were documented, including any clinical fragility fractures by medical record or morphometric vertebral fractures. Traumatic fractures and drug-related atypical fractures were excluded. Due to the high prevalence of asymptomatic vertebral fractures in GIOP [11], all participants underwent an image survey at baseline and 3 years apart to detect any morphometric change in the vertebral bodies, interpreted by independent radiologists according to the Genant semiquantitative assessment method [23]. Secondary osteoporosis was documented, including type I diabetes, osteogenesis imperfecta in adults, untreated long-standing hyperthyroidism, hypogonadism, or premature menopause (younger than 45 years), chronic malnutrition/malabsorption, and chronic liver disease, such as hepatis B or C (https://frax.shef.ac.uk/FRAX/, accessed during September 2014 and April 2021). FRAX was adjusted by GC dosage (GC-FRAX) [24]; that is, for low-dosage exposure (<2.5 mg daily prednisolone or equivalent dosage), the major fracture probability decreased by 20%, with no adjustment for medium dosages (2.5~7.5 mg), and increased by 15% for high-dosage exposure (>7.5 mg daily). Candidate fracture predictors were selected from currently available tools, including the FRAX tool (age, sex, BMI, cigarette/alcohol usage, BMD, secondary diseases, and previous fracture), Garven model (www.fractureriskcalculator.com) (sex, age, fall, and previous fracture), and Q-Fracture tool [25] (age, sex, alcohol/cigarette usage, and BMI). Other variables included ESR, CRP, hemogram, disease duration, and daily steroid dose/duration. Falls were identified through individual recall and self-reported within the preceding year, following the WHO definition, which entailed a person unintentionally coming to rest on the ground or a lower level [26]. Each participant provided written informed consent, and the study was conducted with the approval of the Regional Ethical Review Board of CGMHK (106-0047C).

Study participants were randomly subcategorized with a 2:1 ratio into a derivation and a validation set. All covariables were assessed by logistic regression in the derivation cohort, and the β-coefficients of potential variables were subsequently analyzed with bootstrap replication to determine the weight of the variable, thus formulating the Steroid-Associated Fracture Evaluation (SAFE) tool. The performance of the developed tool was later confirmed in the validation cohort and compared with treatment recommendations by the ACR, IOF-ECTS, and GC-FRAX above the intervention threshold.

### 2.2. Current Treatment Decision Tool for GIOP and Comparison

Current treatment recommendations are summarized in Appendix A. According to ACR recommendations (2017 and 2022 update) [20,27], AOT is recommended for moderate, high, or very high risk patients, who are defined as adults over 40 years of age with prior osteoporotic fractures, a hip or spine BMD T-score below −1, a GC-FRAX score above 10% for major osteoporotic fractures or over 1% for hip fractures, or with very high doses of GCs (≥30 mg/day for >30 days or a cumulative dose of ≥5 g/year) [20]. IOF-ECTS recommended AOT in patients over 70 years old or with a previous fragility fracture, for those under high dosage prednisolone or equivalent doses, or those above country-specific intervention thresholds. Currently, no universal intervention threshold has been established for GIOP in Taiwan; we adopted the clinical practice guideline from the Taiwan Osteoporosis Association, which suggests pharmacotherapy for those with FRAX-major osteoporotic fractures over 20% or hip fractures over 3% [28].

### 2.3. Statistics

Descriptive analysis was presented as means with standard deviation or frequency with percentage. Means between two independent groups were examined using Student’s *t*-test, whereas the chi-square test evaluated categorical variables. All statistical analyses were considered significant for *p* values < 0.05 (α = 5%). In the univariate analysis, we selected candidate predictors with a prevalence of at least 5% and a *p*-value ≤ 0.20. Then, we proceeded to the least absolute shrinkage and selection operator (LASSO) analysis, which reduced the data dimensionality and identified optimal predictors [29,30]. Nonzero coefficients in the LASSO regression were selected for a multiple variable analysis according to the number of predictors confined by one standard error from the minimum lambda value. The β-coefficient obtained from a multiple variable logistic regression was tested with 1000 bootstrapping replications, and candidate variables with replications of over 60% were selected to formulate the model [31]. The bootstrapping method was applied for internal validation to avoid over-optimization and reduce overfit bias. The scores in the SAFE tool were determined by the weight of the β-coefficient of 1000 bootstrapping replications. We illustrated the receiver operating characteristic (ROC) curve to assess the predictive performance among different guidelines, and it was summarized as an area under the curve (AUC) with a 95% confidence interval (CI). The sensitivity, specificity, positive and negative predictive values, and accuracy were calculated for comparison. The bootstrapping method and LASSO regression were calculated and illustrated via R software (R Development Core Team, 2020). Other statistical analyses were performed using the Statistical Package for the Social Sciences (SPSS, version 20.0, Chicago, IL, USA).

## 3. Results

### 3.1. Study Participant Selection and Characteristics

Among the 610 participants in this study, 494 were older than 40 years of age and had been exposed to GCs for over 3 months. After excluding 93 subjects who had been exposed to AOT, the remaining 401 subjects were randomized into the derivation (272, 67.8%) and validation cohorts (129, 32.2%) (Figure 1). The current study showed the mean age was 57.5 ± 8.3 years, with females being the majority (86.0%), and 78 (19.5%) developed new fragility fractures within 3 years (Appendix A). Rheumatoid arthritis (80.8%) was the major reason for long-term GC use, and over 90% of participants had been exposed to GCs for at least 1 year. The average GC cumulative dose was 378.3 ± 249.6 mg within 3 months, or 775.6 ± 417.3 mg within 6 months. The demographic characteristics of the randomized derivation and validation sets were comparable (Appendix A).

### 3.2. Identification of Candidate Predictor Variables Associated with New Fractures

The fracture participants, compared to the non-fracture ones, were featured by an older age (60.0 ± 8.8 vs. 56.9 ± 8.3, *p* = 0.02), a higher previous fracture rate (38.8% vs. 17.9%, *p* < 0.01), a higher osteoporosis prevalence (42.9% vs. 24.7%, *p* = 0.01), and a lower BMD at all three parts (the femoral neck, total hip, and vertebral spine). Expectedly, the 10-year fracture probabilities determined by the GC-FRAX were significantly higher in the fracture group (23.1 ± 16.2 vs. 15.5 ± 11.0 for major fractures, 11.4 ± 13.2 vs. 6.1 ± 7.4 for hip fractures, both *p* < 0.01). The incidence rate of falls was higher in the fractured group than in the non-fractured group without reaching statistical significance (26.5% vs. 17.0%, *p* = 0.12). We present the difference between the fracture and non-fracture groups of the derivation cohort in Table 1. All the variables proceeded to univariable logistic regression analysis (Appendix A). For the convenience of final formula development, we converted the numeric variables to categorical variables for age, BMI, and GC accumulative dose by choosing the optimal cutoff point of each ROC curve. The results of the univariate logistic analysis are shown on the left panel of Table 2. At the same time, a *p*-value < 0.2 proceeded to an analysis by LASSO regression (Appendix A), and seven potential candidates were confined by one standard error from the minimum lambda value, including osteoporosis, a previous fracture, BMI, GC accumulative dose, age, a previous fall, and the GC-FRAX. These selected candidate predictors then entered the multivariable analyses with 1000 bootstraps, demonstrating that the fractured individuals were featured by an age of over 70 years (odds ratio 4.05, 95% confidence interval [1.11–14.82], 60.0% presence with 1000 bootstraps), a BMI > 23.5 (2.22 [1.10–4.25], 81.4%), a GC accumulative dose ≥ 750 mg within 6 months (2.10 [1.03–4.25], 72.5%), a previous fracture rate (2.03 [0.98–4.20], 72.4%), osteoporosis (2.32 [1.12–4.81], 81.5%), and a previous fall (1.53 [0.69–3.39], 40.8%) (Table 2). To screen the optimal prediction model, the candidate predictors were sequentially added according to the weight of the coefficient, and the ROC curve and calibration plot were illustrated (Appendix A). The presence in bootstrap replications of the variable “fall“ was less than 60%, so we did not add this variable to the final model.

### 3.3. Discrimination and Calibration of Steroid-Associated Fracture Evaluation (SAFE) Tool and Diagnostic Performance

The final formula was the sum of five major variables, including previous fracture (1 point if yes), osteoporosis (1 point if yes), BMI ≥ 23.5 (1 point if yes), GC accumulative dose ≥ 750 mg within 6 months (or equivalent prednisolone ≥ 4.5 mg/day for 6 months) (1 point if yes), and age ≥ 70 years (2 points if yes). The SAFE model demonstrated the discrimination of the AUC = 0.71 (95% confidence interval 0.64–0.79) and Hosmer–Lemeshow test χ2 = 8.24; *p* = 0.22 (Appendix A), while the GC-FRAX was 0.65 (0.56–0.74) in the derivation set. In the validation set, the ROC of the SAFE tool and GC-FRAX were 0.63 (0.51–0.75) and 0.65 (0.54–0.76), respectively. The calibration curve showed little deviation from the ideal line, presenting a slope = 0.966, an intercept of 0.00646 in the derivation set, a slope = 0.818, and an intercept = 0.0769 in the validation set (Figure 2). Figure 3 shows the correlation between the predicted fracture risk (by the derivation set) and observed fracture risk (by the validation set). A score of ≥2 points, equivalent to a fracture risk of ≥20%, was considered the appropriate treatment threshold. We compared the treatment-decision performance of the SAFE tool (≥2 points) to the ACR, IOF-ECTS, and GC-FRAX above the intervention threshold, which was defined as a major osteoporotic fracture ≥ 20%, or a hip fracture ≥ 3% [28]. Figure 4 demonstrates the predictive capabilities among various treatment recommendations. In the validation set, the SAFE tool yielded an AUC of 0.65 (95% confidence interval 0.54–0.76, *p*-value = 0.02), while the ACR 0.56 (0.44–0.67, *p* = 0.37), IOF-ECTS 0.61 (0.50–0.72, *p* = 0.07), and GC-FRAX 0.62 (0.52–0.73, *p* = 0.04). The ROC curve of the SAFE tool was comparative to the ACR (*p* = 0.24), IOF-ECTS (*p* = 0.62), and GC-FRAX (*p* = 0.75). The SAFE tool exhibited moderate sensitivity and higher specificity and accuracy (75.9/54.0/58.9), while for the others, the ACR (100.0/11.0/31.0), IOF-ECTS (75.9/46.0/52.7), and GC-FRAX (82.8/42.0/51.2). Similar findings and trends were also observed in the derivation cohort (Figure 4, left panel).

## 4. Discussion

This research provided real-world evidence to validate current treatment recommendations for GIOP, furnishing clinicians with a guide for patient selection. The treatment threshold of the ACR was set low, leading to increased sensitivity but reduced specificity and accuracy in candidate patient selection. While implementing such a strategy can ensure that the most vulnerable individuals receive treatment, it may also lead to the inefficient allocation of medical resources to those who may not need them. The IOF-ECTS, GC-FRAX, and SAFE tool showed similar prediction effects; however, the SAFE tool was more likely to identify participants genuinely prone to fractures, which would help minimize medical resource wastage, due to its increased specificity and positive predictive value. The results were crucial in identifying the vulnerable population while considering the limited budget for pharmacological treatment coverage. In addition, the current treatment-decision tools for GIOP have AUCs that fall below 0.7, indicating inadequate discrimination performance in detecting patients who need treatment. This suggests that there is much room for improvement in refining clinical tools for GIOP. The current treatment recommendations have a ceiling effect, which limits their ability to identify individuals to the maximum level. The limited number of independent variables available to predict GIOP fractures may be the primary reason for this restriction. Conventional risk factors, such as prior fracture, age, or BMD remained the leading causes of GIOP. The addition of GC-associated fracture risks factors, such as GC exposure duration, daily dose, or accumulative dose, may increase the prediction capacity in addition to conventional risk factors. In the ACR (2022) recommendation, GC cumulative dose was incorporated into the criteria, and a cumulative dose ≥ 5 g in the past year was considered a very high risk of fractures. Moreover, introducing novel, decisive variables, like bone markers or a trabecular bone score, may benefit the improvement of fracture predicting tools [32].

GIOP recommendations have been proposed in the past decade, but the overall adherence to guidelines and the treatment rate have remained relatively low [33], approximately 15% to 24% in Canada [34,35], 12% in Europe [36], and 23.3% in Japan [37]. In Taiwan, only 20.3% of long-term GC users received optimal care for GIOP [38]. Treatment recommendations would better assist clinicians in determining eligible cases and convince the payers that such treatments benefit patients [39]. To our knowledge, the real-world performance of different guidelines had yet to be previously validated. This study aimed to address this gap, providing clinical evidence for potential case finding options.

Fracture risk assessment in GIOP patients, distinct from postmenopausal women, required the additional consideration of GC dosage and duration. Nevertheless, the GC effects might be confounded in human bodies due to underlying diseases for treatment [40]. In early RA, adding GC initially arrests bone loss through its anti-inflammatory effects [41]. Still, chronic GC exposure suppresses osteoblasts and osteoclasts, resulting in a low bone remodeling and a 3–5% BMD decrease per year after the first year [42]. A low-dose GC strategy has become the treatment consensus for autoimmune diseases, based on the results of several clinical trials demonstrating the effects of relieving symptoms, improving physical function, as well as retarding radiographic progression [43]; however, recent research has also indicated that even a low dose of GCs is associated with cardiovascular risks [44], infection, fracture, and gastrointestinal events in real-life tolerability registries [6]. Our study further pointed out that cumulative GC dose, rather than daily dose, is an essential fracture risk factor for patients on GC therapy. The importance of cumulative GC dose was supported by another study regarding GC effects on severe infection, showing that the cumulative GC dose taken in the last 2–3 years also increases the infection risk [45]. Unlike previous studies comparing GC with the placebo group, our study aimed to identify high-fracture-risk patients among those under standard GC therapy in practice. Therefore, the average daily doses were almost the same between the fracture and non-fracture groups, insufficient to reflect the actual impacts on fracture. Therefore, considering both dose and duration, the cumulative GC dose would be a relevant risk factor for fracture risk prediction.

Traditionally, a low BMI was considered an independent risk factor for osteoporotic fracture, whereas a high BMI was considered a protective factor, but recent evidence has shown conflicting results. In a meta-analysis investigating the association between fracture and BMI, a high BMI had a protective role for fragility fractures at the population level. However, after adjusting for BMD, a high BMI was still a risk factor for all osteoporotic fractures [46]. A Finnish prospective cohort study followed-up with for 25-years also indicated that a low or high BMI could be a risk factor for fractures, as well as high mortality [47]. Another Chinese study that included 456,921 participants with a 7.96-year follow-up demonstrated a U-shape relationship between BMI and fracture, with the lowest risk of fracture in overweight participants, and a high risk in obese participants [48]. Likewise, the obese population had higher fracture risks after adjusting for BMD, and abdominal adiposity was associated with a high fracture rate [48]. In a Japanese cohort study followed-up with for 10 years, a U-shaped association between BMI and fractures was also observed, but only in the male population, not female, indicating sex- and ethnicity-dependent differences [49]. In the present study, we also found a high BMI to be an independent risk fracture for GIOP fracture after adjusting for multiple variables, including BMD. It was implicated that other parts of body composition, such as fat tissue or muscle mass, might also contribute to fracture. This could be a potential area of interest for future research in GIOP. Moreover, our study revealed that GIOP participants with a BMI over 23.5 (overweight by WHO Asian classification) may have had high fracture risks, while overweight participants were reported as having the lowest fracture risk in the aforementioned studies. This difference might be associated with underlying diseases or long-term steroid use, while other studies focused on the general population. This unique phenomenon might be specific for GIOP, but we may need more data and to recruit participants for further verification.

This study had several strengths. Firstly, the endpoint of fractures included both clinical and morphometric vertebral fractures, confirmed by independent radiologists who compared the initial to the follow-up images of each participant. This method ensured no subclinical or asymptomatic patients would be missed; they may account for over 70% of all incidental vertebral fractures [50]. Secondly, our work used the accumulative GC dose, incorporating dose and duration, to estimate the fracture risk instead of the daily dose, while some guidelines emphasize daily dose [21,27]. However, using a daily dose cannot reflect the actual GC exposure since the daily dose difference is subtle among those under long-term GC exposure, unable to display the actual impacts of GC. Lastly, the study took previous falls into account for fracture risk evaluation. Although this factor was significantly higher in the fracture group, the statistical significance diminished after a multivariable adjustment. This could be associated with the relatively younger-age participants (57.5 ± 8.3 years) in the current study, while falls were conventionally regarded as a fracture risk among older adults [51].

We had some limitations in the study design. Firstly, the participants were followed-up with for only 3 years, and the fracture rate may have been undervalued in a brief observation period. However, recent studies have revealed that patients with short-term high risk, or imminent risk, are associated with fractures within 1 to 2 years afterwards [52], implying that the short-term fracture risk also determines long-term risk [53]. Secondly, this study could not compare participants’ prognoses with AOT because the reimbursement policy restricts AOT prescriptions in Taiwan. Therefore, participants on AOT at baseline may potentially have had a recent fracture or had higher osteoporosis risks than those without AOT. Thirdly, this study was performed at a single center, and it was difficult to trace the compliance of GC treatment or determine if patients acquired additional GC from other hospitals or drug stores. Fourthly, the present study did not measure abdominal/subcutaneous fatness, muscle mass, or related parameters, which could also affect fracture risks in GIOP. Finally, multi-center studies with external validation might be warranted to provide solid information for evaluating the feasibility of this tool.

## 5. Conclusions

The SAFE tool is a rapid decision-making tool for GIOP treatment that incorporates o**S**teoporosis, **A**ccumulative GC dosage (prednisolone or an equivalent dose of ≥750 mg/6 months or ≥4.5 mg/day for 6 months), previous **F**racture, **E**lderliness (≥70 years), and BMI (≥23.5). The SAFE tool is easy to remember and has a comparable discriminative performance to other GIOP treatment decision tools such as the ACR, IOF-ECTS, and GC-FRAX, but it enhances specificity and accuracy of fracture.

## Figures and Tables

**Figure 1 diagnostics-14-00452-f001:**
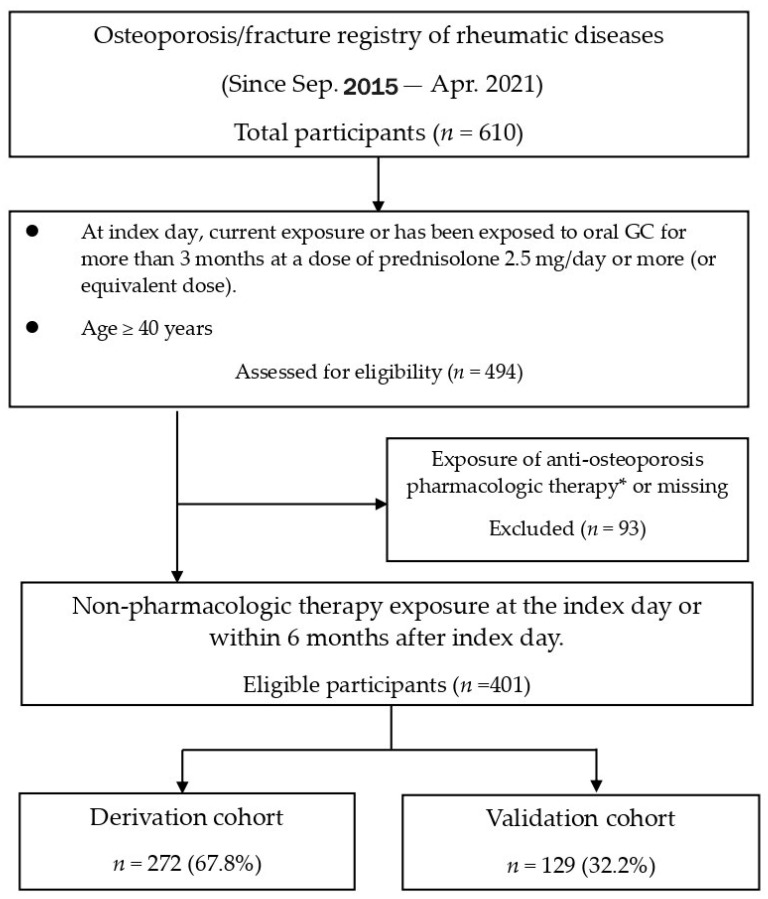
Disposition of participants. GC = glucocorticoid. * Exposure of therapy is defined as therapy exposure currently on the index day, or within the previous 6 months before the index day, or within the upcoming 6 months after the index day. Pharmacologic therapy included selective estrogen receptor modulators, oral or intravenous bisphosphonates, denosumab, and teriparatide.

**Figure 2 diagnostics-14-00452-f002:**
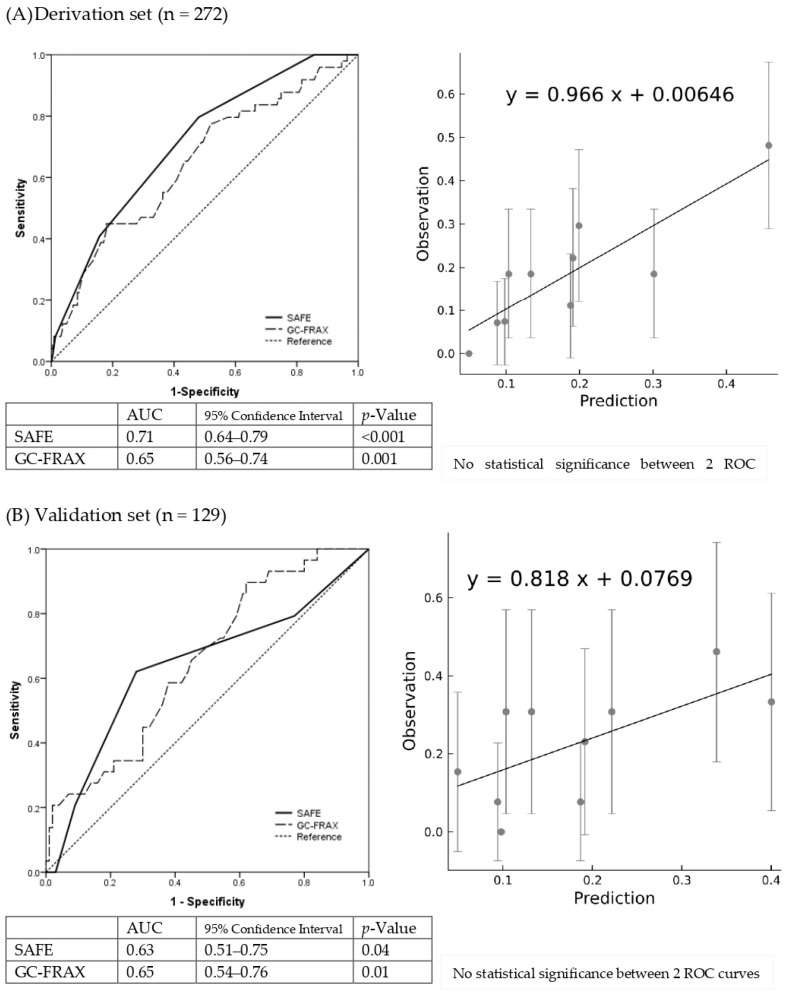
Discriminative and calibration plot of SAFE tool in derivation set (**A**) and validation set (**B**). Discrimination plot (left panel) contains the final model (solid line) in comparison to GC-FRAX (dashed line). Calibration plot (right panel) compares the last models observed and predicted fracture risks.

**Figure 3 diagnostics-14-00452-f003:**
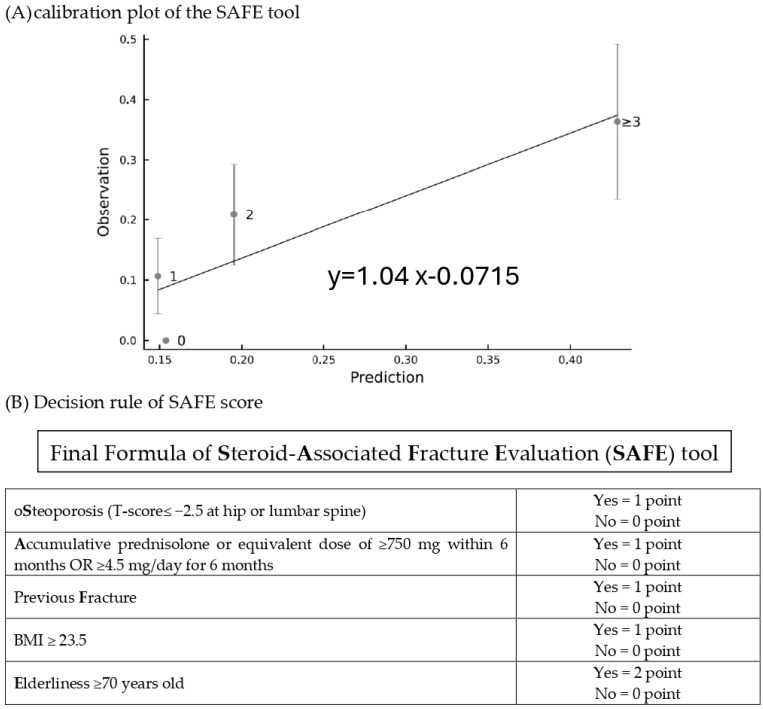
The calibration plot (**A**) of the SAFE tool for prediction risk (derivation set) versus the observed risk (validation set) and decision rules of SAFE score (**B**).

**Figure 4 diagnostics-14-00452-f004:**
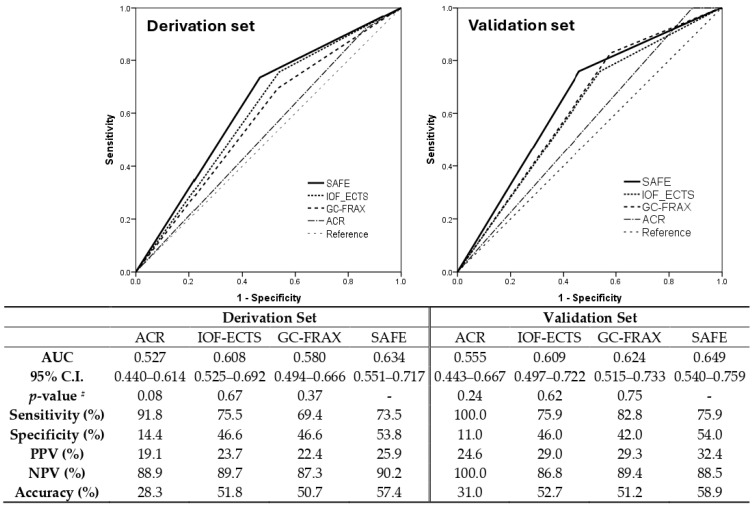
Comparing the performance of treatment recommendation among the SAFE score (≥2 points), ACR, IOF-ECTS, and GC-FRAX (≥intervention threshold). AUC, area under the curve; CI, confidence interval; PPV, positive predictive value; NPV, negative predictive value; ^#^ ROC curve comparison to SAFE tool.

**Table 1 diagnostics-14-00452-t001:** Clinical variables between fracture and non-fracture participants in derivation cohort.

	Derivation Cohort (N = 272)		
Variables	Fracture *n* = 49 (18.0)	Non-Fracture *n* = 223 (82.0)	*p*-Value
Age (years)	60.0 ± 8.8	56.9 ± 8.3	0.02 *
Female, *n* (%)	42 (85.7)	198 (88.8)	0.55
Body weight (kg)	59.4 ± 9.5	58.3 ± 12.1	0.53
Body height (cm)	155.4 ± 6.3	156.4 ± 7.0	0.37
BMI (kgs/cm^2^)	24.7 ± 4.0	23.8 ± 4.1	0.18
Fracture risk factors			
Connective tissue diseases, *n* (%)			
RA	40 (81.6)	183 (82.1)	$\}$ 0.56
SLE	3 (6.1)	23 (10.3)	
Sjögren’s syndrome	2 (4.1)	4 (1.8)	
Others	4 (8.2)	13 (5.8)	
Previous fracture +, *n* (%)	19 (38.8)	40 (17.9)	0.001 *
Osteoporosis +, *n* (%)	21 (42.9)	55 (24.7)	0.01 *
BMD (g/cm^2^)			
FN	0.607 ± 0.117	0.646 ± 0.113	0.03 *
TH	0.759 ± 0.160	0.811 ± 0.138	0.02 *
L1–4	0.820 ± 0.183	0.873 ± 0.162	0.05 *
2nd osteoporosis +, *n* (%)	0 (0)	12 (5.4)	0.10
GC dose (mg/day)			
Daily dose	4.4 ± 1.4	4.1 ± 1.8	0.23
Exposure duration, *n* (%)			
≥3 months	49 (100)	223 (100)	-
≥6 months	45 (91.8)	212 (95.1)	0.37
≥12 months	44 (89.8)	204 (91.5)	0.71
≥2 years	40 (81.6)	179 (80.3)	0.83
≥3 years	36 (73.5)	158 (70.9)	0.71
Accumulative dose			
**Within 3 months**			
<300 mg	23 (46.9)	117 (52.5)	0.48
≥300 mg	26 (40.2)	106 (47.5)	
**Within 6 months**			
<750 mg	15 (30.6)	104 (46.6)	0.04 *
≥750 mg	34 (69.4)	119 (53.4)	
Parent fractured hip +, *n* (%)	4 (8.2)	14 (6.3)	0.63
Current smoking +, *n* (%)	4 (8.2)	10 (4.5)	0.29
Alcohol +, *n* (%)	1 (2.0)	2 (0.9)	0.49
Previous fall +, *n* (%)	13 (26.5)	38 (17.0)	0.12
FRAX			
Major fracture (%)	23.1 ± 16.2	15.5 ± 11.0	<0.01
Hip fracture (%)	11.4 ± 13.2	6.1 ± 7.4	<0.01
Lab			
White blood cell (10^3^/μL)	7.0 ± 2.3	7.1 ± 2.2	0.89
Hemoglobin (g/dL)	12.8 ± 1.5	12.8 ± 1.4	0.84
Platelet (10^3^/μL)	257.6 ± 85.2	251.8 ± 70.5	0.65
BUN (mg/dL)	15.3 ± 3.8	15.0 ± 5.2	0.71
Creatinine (mg/dL)	0.7 ± 0.1	0.7 ± 0.2	0.34
AST (U/L)	25.9 ± 9.6	27.1 ± 16.9	0.66
ALT (U/L)	26.4 ± 15.3	28.5 ± 28.5	0.66
Albumin (g/dL)	4.4 ± 0.3	4.3 ± 0.3	0.28
Calcium (mg/dL)	9.3 ± 0.4	9.3 ± 0.4	0.99
Phosphate (mg/dL)	3.7 ± 0.5	3.7 ± 0.6	0.99

* Fisher’s exact test analyzed *p*-value < 0.05; RA = rheumatoid arthritis; SLE = systemic lupus erythematous; BMD = bone mineral density; FN = femoral neck; TH = total hip; L1–4 = lumbar spine L1–L4; FRAX = Fracture Risk Assessment Tool.

**Table 2 diagnostics-14-00452-t002:** Univariate and multivariate logistic regression analyses in the derivation cohort.

	Univariate Analysis ^%^ with 1000 Bootstrap Replications			Multivariate Analysis ^$^ with 1000 Bootstrap Replications			Presence in Bootstrap Replications, %	Score
	Β-Coefficient (95% CI)	OR (95% CI)	*p*-Value	Β-Coefficient (95%CI)	OR (95% CI)	*p*-Value		
**Age**								
40–50 years		1			1			0
50–70 years	0.55 (−0.29–1.80)	1.73 (0.68–4.35)	0.25	0.28 (−0.58–1.58)	1.32 (0.50–3.52)	0.62		0
≥70 years	1.73 (0.58–3.15)	5.65 (1.70–18.78)	0005 *	1.40 (−0.09–3.12)	4.05 (1.11–14.82)	0.03 *	60.0	2
**BMI**								
<23.5		1			1			0
≥23.5	0.63 (−0.03–1.30)	1.87 (1.00–3.53)	0.05	0.80 (0.13–1.65)	2.22 (1.10–4.50)	0.03 *	81.4	1
**GC accumulative dose**								
Within 3 months								
<300 mg		1						
≥300 mg	0.22 (−0.37–0.88)	1.25 (0.67–2.32)	0.49	-	-	-	-	
Within 6 months								
<750 mg		1			1			0
≥750 mg	0.68 (0.07–1.40)	1.98 (1.02–3.84)	0.04 *	0.74 (0.06–1.67)	2.10 (1.03–4.25)	0.04 *	72.5	1
**Sex**								
Women		1						
Men	0.28 (−0.90–1.11)	1.32(0.54–3.25)	0.55	-	-	-	-	
**Previous fracture**								
No		1			1			0
Yes	1.06 (0.31–1.73)	2.90 (1.48–5.66)	0.002 *	0.71 (−0.16–1.63)	2.03 (0.98–4.20)	0.06	72.4	1
**Osteoporosis**								
No		1			1			0
Yes	0.83 (0.20–1.55)	2.29 (1.21–4.36)	0.01 *	0.84 (0.07–1.75)	2.32 (1.12–4.81)	0.02 *	81.5	1
**Previous fall within 1 year**								
No		1			1			
Yes	0.56 (−0.24–1.29)	1.76 (0.85–3.62)	0.13	0.43 (−0.60–1.31)	1.53 (0.69–3.39)	0.29	40.8	
**GC-FRAX ^&^**								
<IT ^#^		1						0
≥IT ^#^	0.70 (0.07–1.41)	2.02 (1.04–3.91)	0.04 *	-	-	-		1

GC-FRAX = glucocorticoid-dose-adjusted Fracture Risk Assessment Tool; ^&^ GC-FRAX was not conducted in multivariable analysis due to collinearity with other variables. * *p* value < 0.05; ^%^ items were selected for screening only if the *p*-value was less than 0.20 in the derivation cohort. ^$^ items in the multivariable analysis were selected by LASSO logistic regression (see Appendix A), and the B-coefficient of the intercept was −3.32, *p* < 0.001. ^#^ IT, intervention threshold, defined as FRAX major osteoporotic fracture ≥ 20% or hip fracture ≥ 3%.

## Data Availability

The data that support the findings of this study are available from the corresponding author, Tien-Tsai Cheng, upon reasonable request.

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
