# Peer review of "Development and Comparison of Treatment Decision Tools for Glucocorticoid-Induced Osteoporosis"

_diagnostics, 2024, doi:10.3390/diagnostics14040452_

Round 1
Reviewer 1 Report
Comments and Suggestions for Authors
This is a quite interesting paper, aiming to create a treatment decision tool for patients with GIOP and comparing it to other established algorithms. It is clear and well written and the parameters are easy to measure. This tool seems to have a better specificity and accuracy than the treatment decision instruments.
I am a little puzzled by the results regarding high BMI as an independent risk factor for fracture, since it is wellknown that low BMI is such a risk factor in general population. I would like the authors to comment on this issue and to compare their results with other published experiences in this regard.
Reviewer 2 Report
Comments and Suggestions for Authors
Dear Authors,
The manuscript titled: “Development and Comparison of Treatment Decision Tools for Glucocorticoid-Induced Osteoporosis: Comparison of Treatment Decision Tools for GIOP” is written in clear, unambiguous, and professional English, making it accessible to a wide scientific audience. The study is well-grounded in current scientific literature, demonstrating a thorough understanding of the field. The introduction provides an excellent context for the study. The article adheres to a high standard of academic presentation. The results are relevant and directly address the research question. The research topic aligns well with the journal's focus and contributes original findings to the field. A strong aspect of the manuscript is the provision of robust, statistically sound, and well-controlled data. However, while the research question is clear and relevant, the manuscript could have more explicitly stated how this research fills a specific knowledge gap in the field of bone biology. The manuscript's scope is appropriate, and the research holds potential significance in its field. Nonetheless, implications for future research could be added to conclusion. Minor language editing is required. Also, it is important to mention what could be possible clinical relevance to include BMI larger than 23.5 kg/m2 in your formula. Also, it is not clear did you check distinction between abdominal (visceral) adiposity and subcutaneous adiposity as a possible factor in your analysis.
Additional comments are given bellow:
Title: In my opinion it would be better to avoid abbreviations in the title, so please consider removing GIOP from the title. Also, it would be advisable to add study type in abstract and/or title to ease literature search in future studies.
Abstract and Keywords: Please, correct spelling mistakes. We should read “follow-up” rather than “Follow-up” in middle of the sentence. Also, in keywords we should read “Treatment” to be uniform.
Introduction (page 2): Please correct “glucocorticoid” to “glucocorticoids” since not one compound and drug is meant by this term.
Methodology (page )
Results (page 6): . We should read “osteoporosis” rather than “Osteoporosis” in middle of the sentence.
Table 1 (page 6): In my opinion, it would be easier to follow the table if osteoporosis row would be placed closer to other fracture risk factors (preferably closer to secondary osteoporosis row) given that in provided context in could be associated more with parental data rather then original data of the patients.
Table 1 (page 7): Is bone mineral density measurement available for intertrochanteric region of proximal femora, or just femoral neck and total proximal femora BMD were analyzed?
Table 2 (page 7): From my point of view, age ranges should be provided to declutter the table (“40-50 years” rather than “>40 and <50 years”).
Comments on the Quality of English LanguageMinor English editing is required.
